# Emitter Signal Waveform Classification Based on Autocorrelation and Time-Frequency Analysis

**Zhiyuan Ma [1,2]** 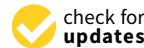 **, Zhi Huang [2], Anni Lin [2] and Guangming Huang [1,***

[1] College of Physical Science and Technology, Central China Normal University, No. 152 Luoyu Road, Wuhan 430079, China; mazhiyuan@mails.ccnu.edu.cn

[2] Department of Electronic Technology, Naval University of Engineering, Wuhan 430033, China; hz5612999@163.com (Z.H.); jiangjiang1821002@163.com (A.L.)

[*] Correspondence: gmhuang@mail.ccnu.edu.cn

**Abstract:** Emitter signal waveform recognition and classification are necessary survival techniques in electronic warfare systems. The emitters use various techniques for power management and complex intra-pulse modulations, which can create what looks like a noisy signal to an intercept receiver, so emitter signal waveform recognition at a low signal-to-noise ratio (SNR) has gained increased attention. In this study, we propose an autocorrelation feature image construction technique (ACFICT) combined with a convolutional neural network (CNN) to maintain the unique feature of each signal, and a structure optimization for CNN input layer called hybrid model is designed to achieve image enhancement of the signal autocorrelation, which is different from using a single image combined with CNN to complete classification. We demonstrate the performance of ACFICT by comparing feature images generated by different signal pre-processing algorithms, and the evaluation indicators are signal recognition rate, image stability degree, and image restoration degree. This paper simulates six types of the signals by combining ACFICT with three types of hybrid model, the simulation results compared with the literature show that the proposed methods not only has a high universality, but also better adapts to waveform recognition at low SNR environment. When the SNR is –6 dB, the overall recognition rate of the method reaches 88%.

**Keywords:** emitter signal waveform recognition; autocorrelation; feature image; hybrid model; low SNR

## 1. Introduction

Electronic warfare (EW) is a military action whose objective is the control of the electromagnetic spectrum (EMS). This objective is achieved through offensive electronic attack (EA), defensive electronic protection (EP), intelligence gathering, and threat recognition electronic warfare support (ES) actions. Electronic intelligence (ELINT) receiver via prolonged and accurate measurement of all the characteristics of a radar emitter (waveform and antenna patterns) in order to provide the necessary data for its analysis and modeling of the associated weapon system, as well as for its identification to be logged in the emitter libraries [1]. In practice, the automatic emitter waveform recognition technique is a core survival technique for an intercept receiver performing threat recognition and radar emitter identification. As the premise and basis of recognition, the emitter signal waveform classification is an important link to ELINT.

In the literature, there have been a few signal waveform recognition techniques that utilize feature extraction techniques and classification techniques to extract features from the intercepted signal, and to classify the intercepted signal based on the extracted features, respectively. To accurately identify emitter signals requires the extraction of features that can represent signals. Time-frequency analysis

(TFA) [2] is widely used in emitter signal waveform recognition and classification. Combined with deep learning in the field of computer vision [3], and models of neural network structures [4,5], researchers have obtained better recognition results from the time-frequency feature of signals [6–8]. However, some problems restrict such developments in environment monitoring, where signals generated by different types of electromagnetic sources are in many situations noisy, misshaped, or changing in relation to the weather condition, task, and application [9,10]. Time-frequency images (TFI) will be polluted at low SNRs, making signals more difficult to classify. To reduce the influence of noise on TFI requires the signal to be processed by an effective filtering algorithm. Some signal-filtering algorithms have focused on one or two dimensions of signal processing. In one-dimensional space, combined with some traditional theory based on the characteristics of random noise with a zero mean [11,12], such as sampling integrals and digital averages, a digital average method to realize low-SNR signal recognition was proposed [8]. With the recent development of the theories of adaptive filtering, autocorrelation detection, and wavelet transforms, more new noise-reduction algorithms have been proposed. Wavelet transforms and autocorrelation detection were combined to improve the detection ability of weak LFM signals [13]. Time-frequency peak filtering (TFPF) was proposed to suppress random noise [14]. Empirical mode decomposition (EMD) and discrete wavelet transforms were combined to reduce noise [15]. In two-dimensional space, noise-reduction algorithms mainly deal with time-frequency matrices. A stacked convolution denoising automatic encoder (SCDAE) was proposed, in which time-frequency data were reconstructed to enhance the signal component in the TFI [7]. A method of image morphology (IM) erosion and expansion to process time-frequency data was proposed [16]. Singular value decomposition was used to decompose the time-frequency matrices of a signal into noise and signal subspaces to reduce the influence of noise [17]. The threshold value of a time-frequency matrix was defined to remove noise as much as possible through threshold filtering [18]. IM and threshold filtering were combined to reduce noise in TFI [19].

However, most noise-reduction algorithms are difficult to apply at a low-SNR environment, such as digital average denoising [11], it did not give further explanation on how to judge the trigger condition of the digital average. In fact, the literature pointed out that finding digital average trigger conditions is also faced with a complex environment with low SNR [8]. For adaptive filtering [12], the noise component completely submerges the signal component in the time-frequency feature image of the signal at low SNRs. In addition, for the two-dimensional noise reduction algorithm, the overall recognition rate of the de-noised signal in [7] is less than 65% in the SNR of –9 dB. The way of reducing time-frequency domain noise by image processing can reduce the noise of TFI [16,18], [19], however, the processing of IM and threshold filtering tends to lose the signal components contained in TFI. The noise-reduction effect for singular value decomposition was simulated at SNR > 0 dB [17], while simulations were not conducted for SNR < 0 dB. After the observation of some denoising algorithms, it can be found that some were merely applied with a certain style of signal, such as LFM [13], seismic signals [14], and electrocardiogram (ECG) signals [15]. However, considering the application of noise-reduction algorithms to specific signals in unknown space, the premise is still to understand the modulation styles of specific signals.

In view of the high requirement of signal de-noising at low SNR and the practical requirement of emitter signal recognition task, our previous studies used autocorrelation images and multiple parallel CNNs to improve signal recognition rates, but lacked theoretical analysis of the causes of autocorrelation images [18]. Since the classification task are different from signal parameter measurements, the techniques provided in this manuscript are intended to meet the requirements for signal classification at low SNR, but the measurement of signal parameters is not considered. Therefore, we propose a signal autocorrelation feature image construction technique (ACFICT) which a signal's autocorrelation sequence is acquired, and the sequences are converted to the two-dimensional space after TFA. Unlike the TFI, which is obtained directly from the TFA, a TFI combined with the ACFICT can maintain the unique feature of each signal. This manuscript further explains the construction techniques of autocorrelation images on a theoretical level. The feature images formed by different

algorithms are comprehensively evaluated according to the three image evaluation indicators in signal recognition rate, image stability degree, and image restoration degree. Then, unlike taking the single CNN to classify, we combine two groups of CNN and one group of bi-directional long short-term memory (BiLSTM) as classifier. For the convenience of description, the structure is called hybrid model in this manuscript. The three groups of models are not connected in the feature extraction stage. Additionally, the input-layer structure optimization is further analyzed by the signal recognition rate and time consumption. We verified the universality of ACFICT by changing the network structure of CNN in the hybrid model. In comparison with the competition literatures, the simulation results further prove that ACFICT combined with the hybrid model is a more effective signal modulation type classification method.

The remainder of this manuscript is organized as follows. Section 2 introduces the basic framework of emitter signal recognition based on deep learning. Section 3 proposes a pre-processing method for signals and the evaluation indices defined by the signal pre-processing and emitter-signal recognition tasks. Section 4 presents the results of a simulation experiment, and the dataset-generation mode of the experiment is given. Section 5 relates our conclusions.

## 2. Basic Recognition Framework

In this section, we present the basic framework of signal classification based on deep learning. The fundamental task of signal classification is to classify various signals captured from the space. The methods based on the deep learning gain a better performance currently [11,20,21]. The specific process to signal is shown in Figure 1, the space signal is captured into the signal receiver via the antenna. The intermediate frequency (IF) sampling signals $y(k)$ will be pre-processed which can decrease the effect of spatial noise on the recognition task. Additionally, traditional pre-processing may be performed in one-dimensional space, or it may be combined with signal transformation to multi-dimensional space and then pre-processed. The authors of [7,19,21] denoise the signal by analysing TFI in the time-frequency transform domain. In [11], the two pre-processing modes are combined. First, digital averaging is performed on $y(k)$, and then the TFI of the signal is processed in the time-frequency transform domain. The ACFICT proposed in this manuscript also combine the two pre-processing procedures together, where $y(k)$ dealt with autocorrelation is converted into TFI that has positive robustness to noise, then the TFI is processed by down sampling to reduce image size and the memory consumption on the graphics processing unit (GPU) and central processing unit (CPU). Finally, the pre-processed signal is input into the pre-designed and pre-trained classifier to classify the signal modulation mode. To stress that this manuscript is similar to [19,22–24], and assume that we sample a complete data in one pulse of the signal.

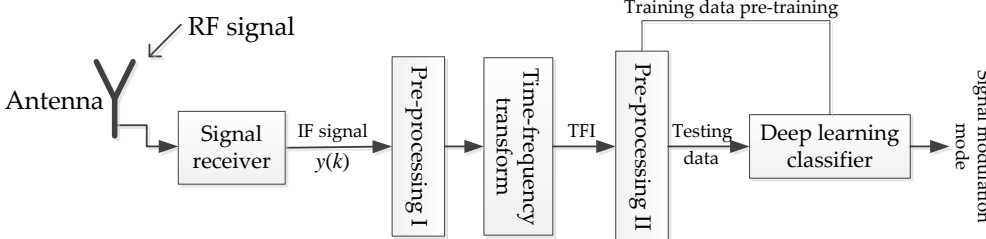

**Figure 1.** The framework of emitter signal waveform classification.

## 3. Propose Signal Pre-Processing Method

In this section, an image-construction algorithm of the signal is proposed to construct a feature image that can describe a signal with high robustness at a low SNR. Combined with autocorrelation detection and time-frequency transform theory, the algorithm can produce a feature image and achieve image enhancement. To evaluate the performance between the proposed algorithm and other algorithms, considering different signal classification methods based on images, they all have similar

requirements for signal feature images, this manuscript proposes three evaluation indices: Stability, recoverability, and recognition rate.

### 3.1. Signal Pre-Processing

#### 3.1.1. Emitter Signals

The signal from the receiving module can be generally expressed as follows:

$$y(k) = x(k) + n(k) = A(k)e^{j\theta(k)} + n(k) \tag{1}$$

where, $x(k)$ is the ideal discrete signal coming after if sampling, $n(k)$ is additive white Gaussian noise, $k$ is the index value increasing with sampling interval sequentially, $A(k)$ is the instantaneous envelope of the ideal sampling signal, $\theta(k)$ is the instantaneous phase of the ideal sampling signal. Additionally, instantaneous phase $\theta(k)$ can be computational expressed further by instantaneous frequency $f(k)$ and instantaneous phase offset $\phi(k)$ as follows:

$$\theta(k) = 2\pi f(k)(kT_s) + \phi(k) \tag{2}$$

where, $T_s$ represents the signal sampling interval. In practice, various signals are usually realized by changing the frequency (frequency modulation) and phase (phase modulation) of the signal. In this manuscript, six types of signals are simulated to verify the algorithm which are conventional phase (CP), linear frequency modulation (LFM), nonlinear cosine phase modulation (NCPM), binary phase shift keying (BPSK), binary frequency shift keying (BFSK), and quadrature phase shift keying (QFSK). Some detailed information such as instantaneous envelope, instantaneous frequency, and instantaneous phase offset of the six types above are shown in Table 1.

**Table 1.** Parameters of signal.

| Modulation Mode | $f(k)$ [Hz] | $\phi(k)$ [rad] | $A(k)$ |
|---|---|---|---|
| CP | $f_c$ | constant | 1 |
| LFM | $f_c + B(kT_s)/\tau_{pw}$ | constant | 1 |
| NCPM | $f_c$ | $f_c - f_k \sin 2\pi f_k(kT_s)$ | 1 |
| BPSK | $f_c$ | 0 or $\pi$ | 1 |
| QFSK | $f_1, f_2, f_3, f_4$ | constant | 1 |
| BFSK | $f_1, f_2$ | constant | 1 |

As shown in Table 1, the carrier frequencies $f_c$, $f_1$, $f_2$, $f_3$, and $f_4$ are fixed values. In the modulation of LFM, $B$ is the signal bandwidth, $\tau_{pw}$ is the signal pulse width, $kT_s$ is the discrete sampling time. In the modulation of NCPM, $f_k$ is the modulation frequency of the instantaneous phase.

#### 3.1.2. Time-Frequency Transformation

The transformation used in this manuscript is the Choi–Williams distribution (CWD). When applied to signal $x(t)$, the time-frequency transformation with the bilinear form is expressed as

$$C_x(t, \Omega) = \iint \sqrt{\frac{\pi\sigma}{\tau^2}} x(u + \tau/2)x^*(u - \tau/2)\, e^{-\pi^2\sigma(u-t)^2/4\tau^2 - j\Omega\tau}\, du d\tau \tag{3}$$

where $C_x(t, \Omega)$ is the result of TFA, $t$ and $\Omega$ are the time and frequency axis, respectively, and the scale factor $\sigma$ is used to control the distribution of CWD cross-terms. When $\sigma$ inhibits the cross-terms in CWD, the frequency resolution decreases. The value $\sigma = 1$ is used to balance the relationship between CWD cross-term suppression and the frequency resolution.

Six types of signals are simulated with an SNR of 9 dB in this manuscript. The TFIs of the six signals that can be obtained by TFA are shown in Figure 2.

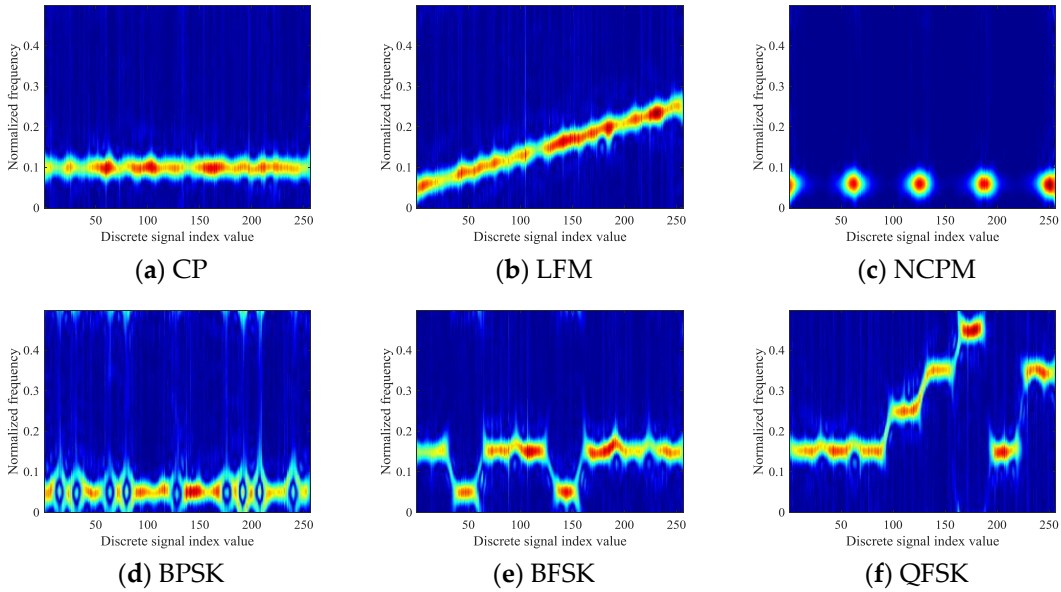

**Figure 2.** Time-frequency images of six types of signals: (**a**) Conventional phase (CP); (**b**) linear frequency modulation (LFM); (**c**) nonlinear cosine phase modulation (NCPM); (**d**) binary phase shift keying (BPSK); (**e**) binary frequency shift keying (BFSK); (**f**) quadrature phase shift keying (QFSK).

### 3.1.3. Signal Feature Images Construction

Given a signal $x(t)$ and zero-mean Gaussian white noise $n(t)$, the observable signal is expressed as $y(t) = x(t) + n(t)$, So, the autocorrelation is simplified to

$$\begin{aligned} R_y(\tau) &= E[y(t)y(t-\tau)] = E\{[x(t)+n(t)][x(t-\tau)+n(t-\tau)]\} \\ &= E[x(t)x(t-\tau)] + E[n(t)n(t-\tau)] + E[x(t)n(t-\tau)] + E[n(t)x(t-\tau)] \\ &= R_x(\tau) + R_n(\tau) + R_{xn}(\tau) + R_{nx}(\tau) \end{aligned} \quad (4)$$

Since the noise $n(t)$ is not related to the signal $x(t)$, we can conclude that $R_{xn}(\tau) = R_{nx}(\tau) = 0$, and then $R_y(\tau) = R_x(\tau) + R_n(\tau)$. For zero-mean Gaussian white noise $n(t)$ with wide bandwidth, its autocorrelation function $R_n(\tau)$ mainly affects the nearby position $\tau = 0$, and when the value of $\tau$ is large, then $R_x(\tau)$ can be characterized approximately by $R_y(\tau)$, that is, $R_x(\tau) \approx R_y(\tau)$. For convenience, we will write $R_y(\tau)$ as $R_y(t)$. Then, combined with the TFA result of the autocorrelation function, $R_y(t)$ is calculated after being taken into Equation (3):

$$C_R(t,\Omega) = \iint \sqrt{\frac{\pi\sigma}{\tau^2}} R_y(u+\tau/2) R_y{}^*(u-\tau/2)\, e^{-\pi^2\sigma(u-t)^2/4\tau^2 - j\Omega\tau} du d\tau \quad (5)$$

According to Equation (5), the result obtained by the Choi–Williams transformation reflects the change of the autocorrelation function $R_y(t)$. As $R_y(t)$ is determined only by the signal $x(t)$, the obtained time-frequency transform $C_R(t,\Omega)$ is a response of $x(t)$ in the time-frequency plane after the autocorrelation domain, so it can represent the signal $x(t)$ uniquely.

In the actual signal reception, the IF sampling of $y(t)$ is always the discrete value noted as $y(k)$, and the autocorrelation function of the discrete values is

$$\hat{R}_y(k) = \sum_{n=0}^{N-1} y(n)y(n-k) \quad (6)$$

where $N$ is the number of sampling points, and $\hat{R}_y(k)$ is the estimated value of $R_y(\tau)$. When $n < 0$, then $y(n) = 0$. So, the discrete transformation $\hat{R}_y(k)$ has the relation

$$C_R(i,l) = \sum_m \sum_k \sqrt{\frac{\pi\sigma}{m^2}} \hat{R}_y(k+m)\hat{R}_y^*(k-m)\, e^{-\pi^2\sigma(k-i)^2/16m^2 - j4\pi ml/N} \tag{7}$$

Then, $C_R(i,l)$ will be inverted into the pixel range considering the need to convert $C_R(i,l)$ to a picture, and the pixel value range of the image is 0 ~ 255.

For convenience of expression, $C_{ij}$ represents the data in the two-dimensional matrix $C_R(i,l)$ of row $i$ and column $j$, $C_R$ represents $C_R(i,l)$, and the equation can be written as

$$Picture = 255\frac{C_{ij} - \min(C_R)}{\max(C_R) - \min(C_R)} \tag{8}$$

where $\max(\bullet)$ and $\min(\bullet)$ respectively represent the maximum and minimum points of the two-dimensional matrix, and *Picture* is the feature image of various signals constructed in this manuscript that can represent the image formed by mapping the matrix $C_R(i,l)$ to the pixel interval. Finally, the feature image is compressed to size $64 \times 64$ by extracting pixels at equal intervals to reduce the computational load of the CPU and GPU.

Combined with signal autocorrelation and TFA, the autocorrelation images of the six types of signals are obtained as shown in Figure 3 at an SNR of 9 dB. It can be seen that the noise appearing in Figure 2 disappears significantly after the signals have been autocorrelation processed.

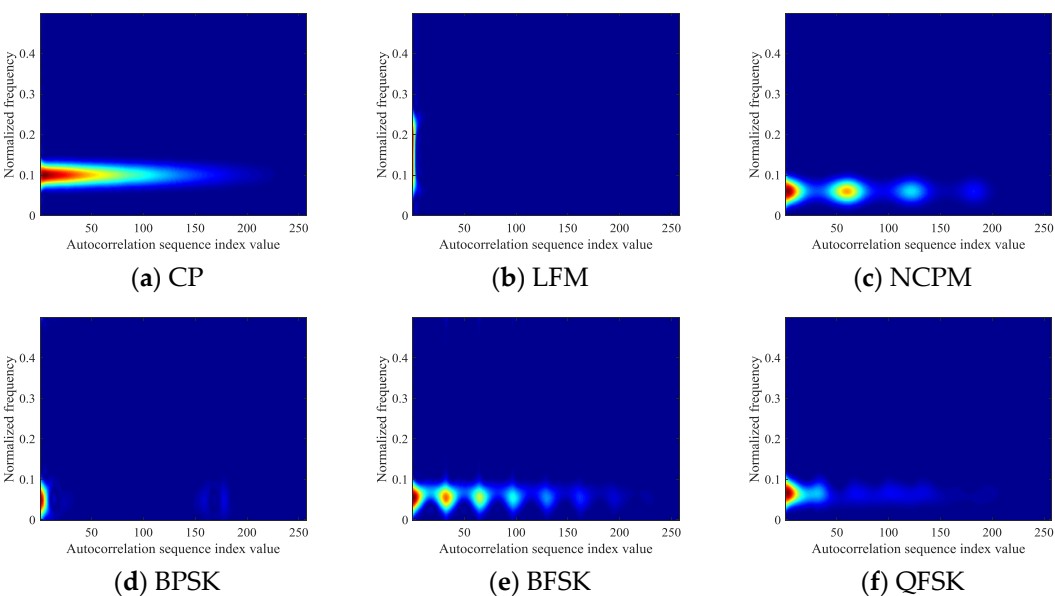

**Figure 3.** Autocorrelation images of signals: (**a**) CP; (**b**) LFM; (**c**) NCPM; (**d**) BPSK; (**e**) BFSK; (**f**) QFSK.

According to Equation (6), the autocorrelation value $\hat{R}_y(0)$ reaches the maximum when $k = 0$ and decreases as $k$ increases. The case found from the observation of Figure 3 is that the autocorrelation images of the signals have the largest pixel value at the left edge, while the pixel gradually weakens and disappears along the right side, conforming to the distribution rule of autocorrelation values in Equation (6). However, due to the effect of autocorrelation calculation, the problem is that we can't extract more features from every pixel in the image, while only few areas of the image have useful information. To solve this problem, the value range of $k$ during the construction of signal feature images is analyzed in Section 3.1.4 based on the distribution law of signal autocorrelation values.

### 3.1.4. Signal Autocorrelation Deviation Analysis

In reality, $\hat{R}_y(k)$ is used as an estimate of $R_y(\tau)$ because the signal after sampling is composed of one-dimensional vectors of finite length.

We further discuss the estimation deviation, which occurs when calculating $\hat{R}_y(k)$, as this will affect the pixel distribution of the signal autocorrelation image. We first take the continuous signal $y(t)$ as an example to evaluate the estimation deviation $\hat{R}_y(k)$. The calculation is usually taken within the limited time $T$, so the autocorrelation function can be expressed as

$$\hat{R}_y(\tau) = \frac{1}{T} \int_0^T y(t)y(t - \tau)dt \tag{9}$$

where $\hat{R}_y(\tau)$ is the estimated value of $R_y(\tau)$ within the finite time, and $\hat{R}_y(k)$ can be obtained after sampling within a time range of integration.

The mean square deviation of $\hat{R}_y(\tau)$ can be expressed as

$$\mathrm{var}[\hat{R}_y(\tau)] = E[(\hat{R}_y(\tau) - R_y(\tau))^2] \tag{10}$$

When the zero-mean white noise $n(t)$ in the received signal $y(t)$ has a Gaussian distribution with bandwidth of $B$, then the variance of $\hat{R}_y(\tau)$ can be calculated as [22].

$$\mathrm{var}[\hat{R}_y(\tau)] \approx \frac{1}{2BT}[\hat{R}_y^2(0) + \hat{R}_y^2(\tau)] \tag{11}$$

When $\hat{R}_y(\tau) \neq 0$, then the normalized root mean square deviation of $\hat{R}_y(\tau)$ can be expressed as

$$\varepsilon(\tau) = \frac{\sqrt{\mathrm{var}[\hat{R}_y(\tau)]}}{\hat{R}_y(\tau)} = \frac{1}{\sqrt{2BT}} \frac{\sqrt{1 + \rho_y^2(\tau)}}{\rho_y(\tau)} \tag{12}$$

After analyzing the deviation between the estimated value $\hat{R}_y(\tau)$ and the theoretical value $R_y(\tau)$ based on Equation (12), it can be found that the normalized deviation $\varepsilon(\tau)$ is jointly determined by signal bandwidth $B$, integral time $T$, and signal autocorrelation coefficient $\rho_y(\tau)$. For the fact that $\hat{R}_y(k)$ can be obtained by sampling and autocorrelation within the integral time, so the deviation of $\hat{R}_y(k)$ should have the expression similar to Equation (12). In fact, according to the Nyquist sampling theorem, the sampling rate of the signal should be greater than twice its maximum angular frequency, that is, $f_s \geq 2B$, where $f_s$ is the signal sampling rate. The number of sampling points $N$ of the integration time $T$ must obey the following expression to maintain information integrity in the continuous-to-discrete transformation, where $T_s$ is the sampling interval time:

$$N = \frac{T}{T_s} = f_s T \geq 2BT \tag{13}$$

When $f_s$ obeys the Nyquist Sampling theorem, the deviation of $\hat{R}_y(k)$ (i.e., $\varepsilon(k)$) is calculated as

$$\varepsilon(k) = \frac{1}{\sqrt{N}} \frac{\sqrt{1 + \rho_y^2(k)}}{\rho_y(k)} \tag{14}$$

where $\rho_y(k) = \hat{R}_y(k)/\hat{R}_y(0)$. The deviation of $\hat{R}_y(k)$ can be estimated reasonably using Equation (14). Different deviations $\varepsilon(k)$ can be obtained based on the autocorrelation coefficient $\rho_y(k)$ and number of sampling points $N$. The value of $\varepsilon(k)$ changes with $k$. Six types of signals are simulated under parameters satisfying Equation (13), with an SNR of 9 dB, to analyze the value and change trend of $\varepsilon(k)$. The change curves of $\varepsilon(k)$ are shown in Figure 4.

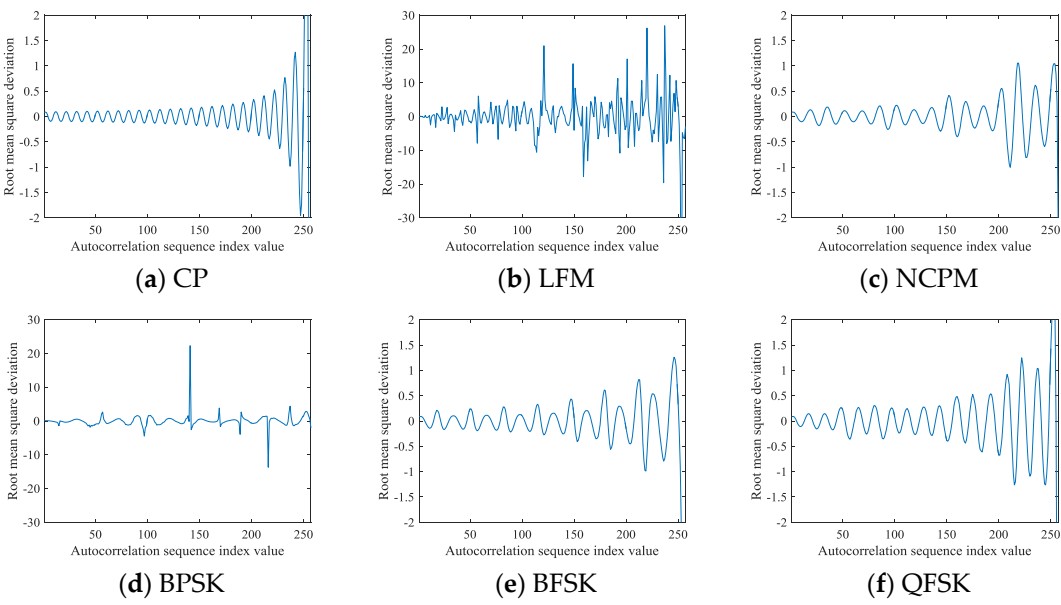

**Figure 4.** The signal $\varepsilon(k)$ curves obtained from Equations (6) and (14): (**a**) CP; (**b**) LFM; (**c**) NCPM; (**d**) BPSK; (**e**) BFSK; (**f**) QFSK.

The number of signal sampling points in Figure 4 is set to 256. It can be found that there are relatively mild changes in $\varepsilon(k)$ at $k \leq 125$, while there will be dramatic deviations in $\hat{R}_y(k)$, which fail to be used to construct the feature images at $k > 125$. In practice, when $k > 125$, the estimation deviation for $\hat{R}_y(k)$ is large because it is calculated from few discrete values, which is the reason why the pixels representing the signal in the autocorrelation images of Figure 3 gradually weaken until they disappear.

In order to obtain more abundant autocorrelation values which can form autocorrelation feature images, we should explore the most appropriate range of $k$ in $\hat{R}_y(k)$. So, the calculation method for signal's autocorrelation result is changed to extend the range of $\hat{R}_y(k)$. Additionally, $\hat{H}_y(k)$ is calculated as

$$\hat{H}_y(k) = \begin{cases} \sum\limits_{n=0}^{N-1} y(n)y(n-k) & k \geq 0 \\ \hat{H}_y(-k) & k < 0 \end{cases} \tag{15}$$

$\hat{R}_y(k)$ is calculated from $\hat{H}_y(k)$ as

$$\hat{R}_y(k) = \hat{H}_y(k-N)k = 1, 2, \ldots, 2N-1 \tag{16}$$

The deviation curves for each signal are updated based on Equation (16) as shown in Figure 5.

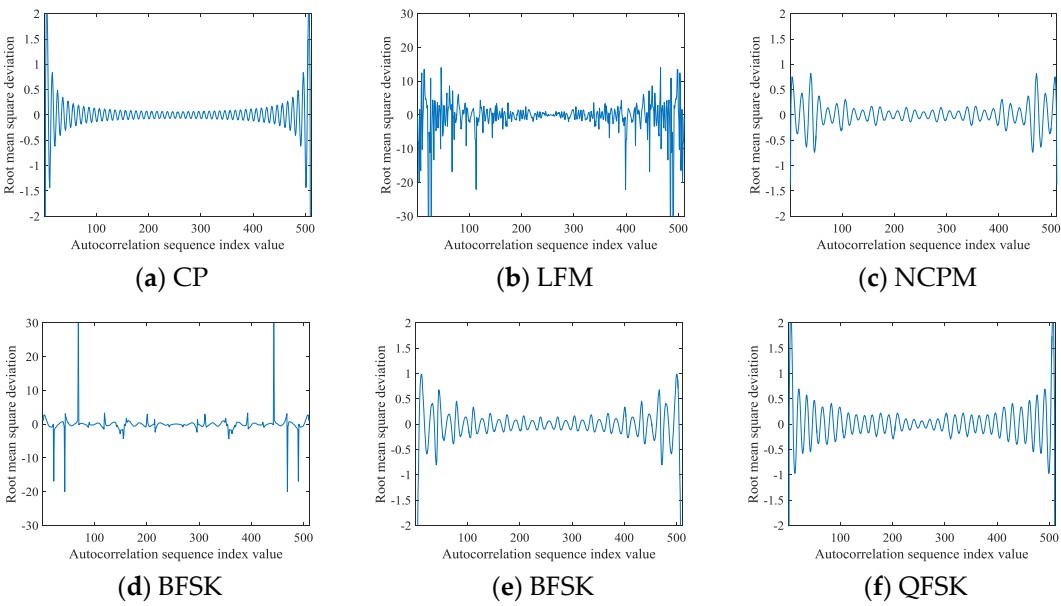

**Figure 5.** The signal $\varepsilon(k)$ curves obtained from Equations (6), (15), and (16): (a) CP; (b) LFM; (c) NCPM; (d) BPSK; (e) BFSK; (f) QFSK.

The number of sampling points in Figure 5 is 511, and it can be found that the deviation value changes slightly when $150 \leq k \leq 350$, but beyond that range, the estimated values $\hat{R}_y(k)$ are calculated from fewer discrete values according to Equations (15) and (16), and thus, the dramatic change will occur again.

After comparing Figure 4 with Figure 5, it can be found that the range $150 \leq k \leq 350$ in Figure 5 has more reasonable autocorrelation values, so it is more suitable for the construction of the signal autocorrelation feature image. We can find that when $\hat{R}_y(k)$ appears to be an unstable change, a large deviation value will appear at the position far from $k = 0$, and the autocorrelation deviation is relatively small near $k = 0$, so this manuscript can further determine the reasonable range of $k$ based on these factors.

For a given sampling signal $y(k)$ with length $N$, an autocorrelation sequence $\hat{R}'_y(k)$ with length $N$ is formed from the range $(N/2, 3N/2)$ of autocorrelation sequence $\hat{R}_y(k)$ with length $2N - 1$, acquired from Equation (16). For convenience, the function $clip(\bullet)$ is used to represent the generation method for the above autocorrelation sequence.

The deviation $\varepsilon(k)$ is constrained in a smaller range and, in turn, the autocorrelation image reconstructed by $\hat{R}'_y(k)$ can more effectively characterize the signal.

We modify the autocorrelation results based on the analysis in this section, and $\hat{R}'_y(k)$ is taken into Equation (6) to construct an autocorrelation feature image with high robustness toward the noise effect. The autocorrelation feature image formed by $\hat{R}'_y(k)$ at an SNR of 9 dB is shown in Figure 6.

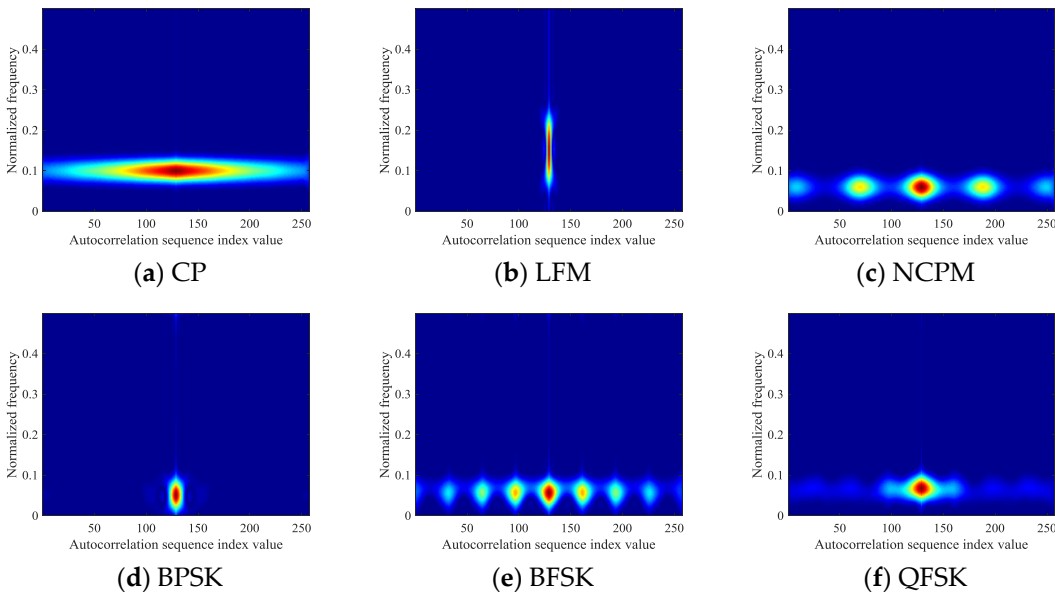

**Figure 6.** Autocorrelation images of signals formed by *clip*(•): (**a**) CP; (**b**) LFM; (**c**) NCPM; (**d**) BPSK; (**e**) BFSK; (**f**) QFSK.

Comparing Figures 3 and 6, it can be seen that extending the range of *k* is efficient for the images in Figure 6, the images have better modification in resolution than those in Figure 3.

In this study, the autocorrelation sequence is formed by intercepting the autocorrelation value of the interval $(N/2, 3N/2)$, which avoids the part of the autocorrelation sequence with large deviation values, so that autocorrelation images can fully reflect the overall change of the signal after the autocorrelation domain, and at the same time, remove the invalid part of the image caused by the larger deviation.

### 3.1.5. Signal Feature Image Enhancement

In this manuscript, the feature image based on autocorrelation theory can effectively represent six types of signals, but its image is still affected by low SNR. Specifically, the pixel strength of the image begins to weaken at low SNR. Figure 7 shows the changes of autocorrelation images of NCPM signals in different SNR environments. It can be found that with the decrease of SNR, the image of NCPM tends to blur. Therefore, in order to obtain feature images that can adapt to low SNR environments, this section is devoted to image enhancement of autocorrelation images of signals.

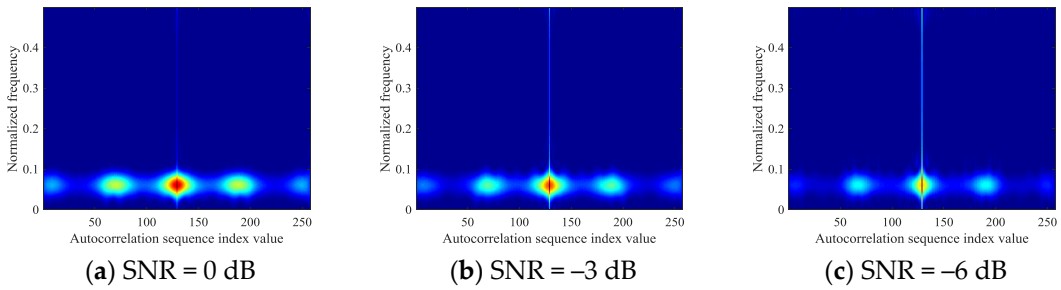

**Figure 7.** The change of NCPM autocorrelation image at different SNR.

In order to achieve image enhancement, the pixels representing the signal should have larger pixel values. The pixel values in this manuscript are the reflection of the signal in time-frequency plane after passing through the autocorrelation domain. If we want to enhance the pixels representing the signal, we should consider improving the autocorrelation values which can represent the signal.

Considering that the iteration calculation of autocorrelation can improve the autocorrelation value of signals, this manuscript proposes to implement image enhancement by multiple autocorrelation of signals. For convenience, the result of $m$ times autocorrelation is expressed as $\hat{H}_y^m(k)$ and $\hat{R}_y^m(k)$, the length of $\hat{R}_y^m(k)$ is noted as $N_m$, and the primitive signal $y(k)$ is noted as $y_0(k)$, the length of $y_0(k)$ is noted as $N_0$. If $\hat{R}_y^m(k)$ is considered as a new signal $y_m(k)$, the autocorrelation result of the sequence $y_m(k)$ is given below.

$$\hat{H}_{y_m}^{m+1}(k) = \begin{cases} \sum_{n=0}^{N_m-1} y_m(n)y_m(n-k) & k \geq 0 \\ \hat{H}_{y_m}^{m+1}(-k) & k < 0 \end{cases} \tag{17}$$

$$\hat{R}_{y_m}^{m+1}(k) = \hat{H}_{y_m}^{m+1}(k-N_m) \ k = 1, 2, \ldots, 2N_m - 1 \tag{18}$$

$$y_{m+1}(k) = \hat{R}_{y_m}^{m+1}(k) \tag{19}$$

Through the iterative calculation of Equations (17)~(19), the autocorrelation result $\hat{R}_y^m(k)$ of the signal after $m$ times autocorrelation can be obtained. However, considering the change of the length of the autocorrelation sequence by Equation (18), iterative calculation will make the sequence length rapid grow, so this section also uses the $clip(\bullet)$ function proposed in Section 3.1.4 to process multiple autocorrelation sequences, so that the length of multiple autocorrelation is also $N$. Based on the image-enhancement theory in this study, the two times autocorrelation image of the signal is shown in Figure 8. Comparing this with Figure 7, it can be found that the feature image of the signal obtained through two times autocorrelation can extract more features from every pixel in the image.

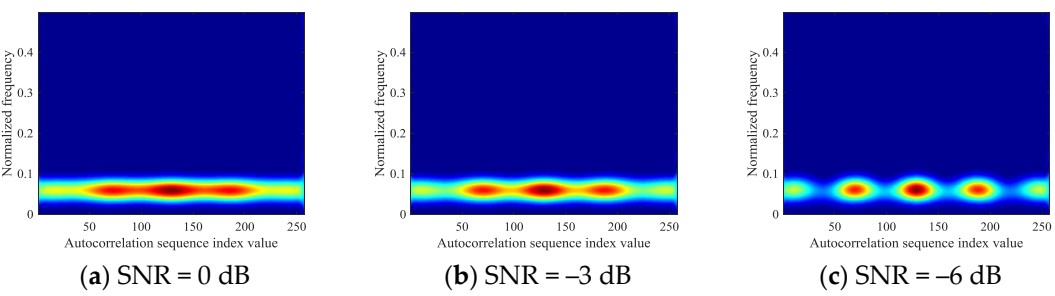

**Figure 8.** The change of NCPM two times autocorrelation image at different SNR.

### 3.2. Model Input Layer Optimization

According to Section 3.1.5, multiple autocorrelation can achieve image enhancement of the signal autocorrelation image, and each autocorrelation can obtain an autocorrelation image that characterizes the signal. In fact, three autocorrelation images characterizing the signal can be obtained by TFA after three times autocorrelation of the signal. More primitive information can be provided if the underlying information of autocorrelation image is used wisely, so we propose an optimized usage for the multiple images as an input of the CNN classification.

As shown in Figure 9, the current TFA-based classification network only takes a single channel TFI as input [7,11,19,21,23], because TFI acquired by TFA is usually a single channel image. However, this manuscript obtains three feature images which can represent signals by iterating the autocorrelation value of signals, which is different from using a single image combined with CNN to complete classification. It means that this method provides more initial information, but the network structure also needs to be further improved. As a result, a deep learning network is designed as shown in Figure 9. In this network, we combine two groups of CNN and one group of BiLSTM to realize the feature extraction of three feature images, and finally complete the signal classification. The three groups of models are not connected in the feature extraction stage. Only after extracting the high-dimensional features, we use the full connection layer to synthesize all the neuron information of each model for classification. For ease of explanation, the network structure is called the hybrid model.

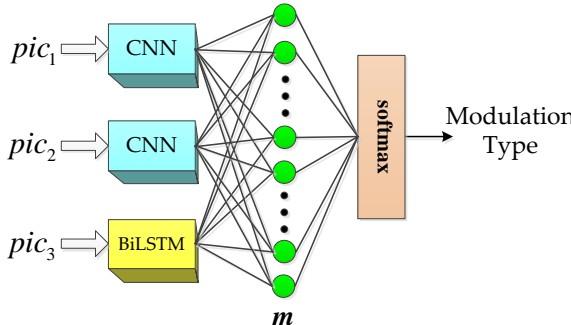

**Figure 9.** Structure optimization for CNN input layer.

It should be emphasized that considering the different hyper parameters of CNN and the depth of the network layer have different effects on the classification results, this manuscript discusses the CNN model used in the classification of signal modulation methods [11,19]. At the same time, a deeper CNN structure is proposed to analyze the impact of different networks on the classification of signals, which will be further explained in Section 4.2.2. For the BiLSTM structure. Zachary [25] points out that if the forgetting gate bias is set to one during model initialization, the output explosion or disappearance problem caused by gradient in the process of model transmission at the beginning can be effectively avoided, so this manuscript sets it to one.

### 3.3. Performance Evaluation of Algorithm

The performance of the algorithm in this manuscript is mainly evaluated according to the signal recognition rate, image stability degree, and image restoration degree. The image stability degree and image restoration degree are indices to evaluate different pre-processing algorithms, which are defined in this section. Aditionally, the signal recognition rate is the final evaluation index of the signal classification task. This manuscript compares it with similar competitive literature.

For the image stability degree, $y(k)$ is the signal to be processed, and feature images **A** and **B** are formed from the effects of noise interference **A′** and **B′**, respectively. When feature images **A** and **B** have high similarity, the given pre-processing algorithm can be proven to be slightly influenced by different noises, i.e., it has high stability.

For the image restoration degree, feature images **A** and **C** in Figure 10 are respectively formed with and without noise interference. If **A** and **C** have high similarity, it shows that the signal pre-processing algorithm can effectively suppress the influence of noise and restore the image formed by the signal without noise.

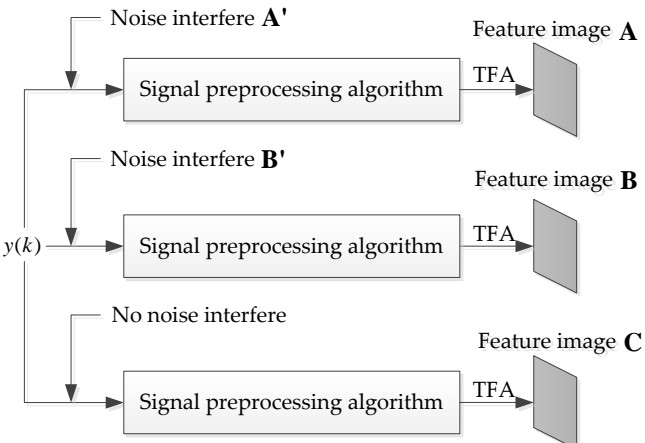

**Figure 10.** Image optimization for CNN input layer.

We use (peak signal-to-noise-ratio) PSNR, which is the logarithmic expression of mean square error, to evaluate the similarity of two images.

For convenience, in feature images **P** and **Q** of size $m \times n$, the pixel points in row $i$ and column $j$ are noted as $\mathbf{P}(i, j)$ and $\mathbf{Q}(i, j)$, respectively, and the mean square error is calculated as

$$MSE = \frac{1}{mn} \sum_{i=1}^{m} \sum_{j=1}^{n} \|\mathbf{P}(i, j) - \mathbf{Q}(i, j)\|^2 \tag{20}$$

PSNR is calculated as

$$PSNR = 10 \log_{10} \left( \frac{255^2}{MSE} \right) \tag{21}$$

According to Equations (20) and (21), the higher the value of PSNR the higher the similarity of the two images. Therefore, a higher value of PSNR indicates a greater image stability degree and image restoration degree.

## 4. Evaluation and Analysis of ACFICT

In this section, the performance of ACFICT is evaluated. We first explain the generation method of the dataset in the simulation experiment. Then, we introduce common pre-processing methods and classification networks. Finally, the proposed algorithm is contrasted with the given pre-processing methods and the classification networks. The performance of ACFICT is evaluated in terms of image stability degree, image restoration degree, signal recognition rate, and time consumption. Additionally, the input-layer structure optimization is further analyzed by the signal recognition rate and time consumption.

### 4.1. Simulation Environment and Data Set Generation

In a real environment, the carrier frequency and signal parameters of various types of signals change all the time. Hence, the values of the parameters of the simulated signals are randomly selected within the specified range in order to keep the simulation consistent with the actual electromagnetic environment.

For a given range $(a, b)$, $U(a, b)$ is used to indicate that the corresponding parameter values are uniformly distributed in the interval from a to b. The parameter value ranges of the six types of emitter signal signals are shown in Table 2 by using the note $U(a, b)$ and the parameters in Table 1.

**Table 2.** Range of signal parameter values.

| Modulation Type | Parameter | Range |
|---|---|---|
| CP | $f_c$ | $U(f_s/6, f_s/4)$ |
| | $\tau_{pw}$ | $U(5e^{-6}, 20e^{-6})$ |
| LFM | $f_c$ | $U(f_s/6, f_s/4)$ |
| | $B$ | $U(f_s/20, f_s/10)$ |
| | $\tau_{pw}$ | $U(5e^{-6}, 7.5e^{-6})$ |
| NCPM | $f_c$ | $U(f_s/6, f_s/4)$ |
| | $\tau_{pw}$ | $U(5e^{-6}, 20e^{-6})$ |
| | $f_k$ | $U(f_s/100, f_s/50)$ |
| BPSK | $f_c$ | $U(f_s/6, f_s/4)$ |
| | $L_c$ | $\{7, 11, 13\}$ |
| BFSK | $f_1$ | $U(f_s/6, f_s/3)$ |
| | $f_2$ | $U(f_s/3, f_s/2)$ |
| | $N$ | $U(256, 512)$ |
| | $L$ | $U(30, 40)$ |
| | $N\_ranf$ | $g(rand(1, 50))$ |
| QFSK | $f_1$ | $U(f_s/6, f_s/4)$ |
| | $f_2$ | $U(f_s/4, f_s/3)$ |
| | $f_3$ | $U(f_s/3, 5f_s/12)$ |
| | $f_4$ | $U(5f_s/12, f_s/2)$ |
| | $N$ | $U(256, 512)$ |
| | $L$ | $U(30, 40)$ |
| | $N\_ranf$ | $h(rand(1, 50))$ |

In Table 2, $f_c$, $f_1$, $f_2$, $f_3$, and $f_4$ are different carrier frequencies; $\tau_{pw}$ is the pulse width; $f_k$ is the modulation frequency of the NCPM signal; $L_c$ is the code length of the BPSK; $N$ is the number of sampling points in the signal; and $L$ is the code length. Within the encoding cycle of a BFSK, two-frequency conversion of a BFSK signal depends on the parameter $N\_ranf$, where $g(rand(1, 50))$ represents that 50 random numbers are generated between 0 and 1 and divided by the boundary of 0.5. Similarly, in the encoding cycle of a QFSK, four-frequency conversion of a QFSK signal depends on the parameter $N\_ranf$, $h(rand(1, 50))$ represents that 50 random numbers are generated between 0 and 1 and divided by the boundary of 0.25, 0.5, and 0.75.

The training and test sets are generated separately, based on the parameter configuration of the six types of signals. The data of the training set are generated from SNRs of –9 to 9 dB, in steps of 3 dB. Each SNR environment has 500 training samples for each type of signal, for a total of 21,000 training samples. Similarly, the data of the test set are generated from SNRs of –9 to 6 dB in 3 dB steps. Every SNR has 300 testing samples for each type of signal, for a total of 12,600 testing samples. The model was built in the Python Tensorflow framework with the Nvidia 1050Ti GPU.

*4.2. Signal Pre-Processing Algorithm and Classification Network Model*

4.2.1. Signal Pre-Processing Algorithm

To reasonably compare the performance differences between various algorithms, the signal pre-processing algorithms commonly used in signal classification are presented. Since the CNN processes two-dimensional data, the CWD transformation mentioned in Section 3.1.2 was applied to various signal pre-processing algorithms to convert one-dimensional data to two-dimensional data to meet the input data requirements of the CNN.

Common signal pre-processing algorithms can be categorized as having one or two processing dimensions. Commonly used one-dimensional algorithms are the adaptive filter [8,26] and wavelet transform (WT) [27]. The literature about [28] in comparison of LMS adaptive filtering algorithm, points

out that the comprehensive performance of the normalized least mean square (NLMS) algorithm is the best. The ACFICT proposed in this manuscript is mainly processed at the one-dimensional level. After the signal is processed by the one-dimensional signal processing algorithm, the data are mapped to the pixel interval to form a feature image that can represent the signal. On the two-dimensional level, the original signal is directly transformed into two-dimensional data by CWD, and the two-dimensional signal processing algorithm is processed on two-dimensional data. The commonly used signal processing algorithm is image threshold denoising (ITD) [19]. Literature [20] proposed combining image morphology with ITD, and the two-dimensional data obtained by signal processing algorithm are mapped to the pixel interval to become the feature image of the signal.

To sum up, we present the signal pre-processing algorithms in Table 3, including LMS, NLMS, wavelet threshold denoising, and the proposed ACFICT at the one-dimensional level. The two-dimensional level includes image threshold denoising and combining ITD with IM. In addition, since CWD can be used as a signal pre-processing algorithm, we consider its performance when used alone.

**Table 3.** Signal pre-processing algorithms.

| No | ① | ② | ③ | ④ | ⑤ | ⑥ | ⑦ |
|---|---|---|---|---|---|---|---|
| Procedure / Dimension | One Dimension | | | | Two Dimensions | | |
| 1 | LMS | NLMS | WT | ACFICT | CWD | | |
| 2 | CWD | | | | ITD | ITD | None |
| 3 | None | | | | None | IM | None |

### 4.2.2. Network Model

Among the different deep-learning classifiers, CNN has been widely used in [7,20,21,23]. Since the performance of the proposed signal pre-processing algorithm on each network model needs to be evaluated reasonably, in order to express the structure of the network model easily, the structure of CNN is modelled in this manuscript. It performs feature extraction and dimensionality reduction of images through alternate convolution and pooling operations, and its classical network structure usually consists of five parts: The input layer, the convolution layer, (i.e., down sampling layer), the fully connected layer, and the output layer.

As Ian G has already given the specific calculation methods of convolution, pooling, and full connection [28], this manuscript will not go into details about their calculation.

The performance of the algorithm in this manuscript was evaluated with three signal-classification networks. In the network model of signal classification, literature [20] and literature [11] both designed the corresponding network structure and achieved a good recognition rate through super-parameter tuning. The network structure is shown in Figures 11 and 12. Considering the development of the current deep network model, we designed the network structure shown in Figure 13. Among the three networks, the network structure in reference [11] (Figure 12) has more convolution kernels and feature maps than Figure 11). The network presented in this study (Figure 13) has deeper network layers than the others. By designing three network structures, we comprehensively evaluated the performance of the algorithm on different network classifiers. It should be emphasized that in the three networks, the unified input image size of this manuscript is $64 \times 64$, the convolution fill is uniformly zero padded, the block size is set to 256, and the learning rate is set to 0.01. For convenience, the network structure of Figure 11 is called CNN1, the network structure of Figure 12 is called CNN2, and the network structure of Figure 13 is called CNN3.

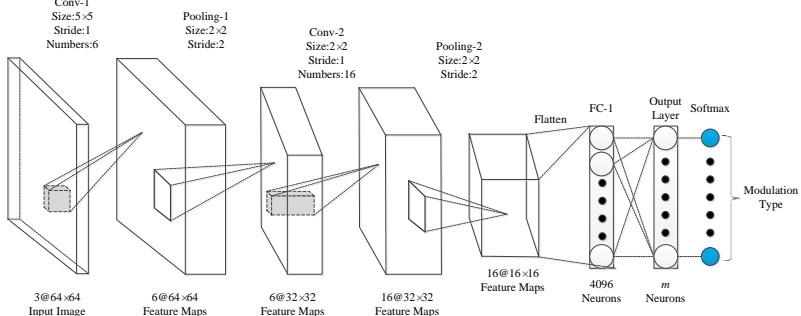

**Figure 11.** Network structure diagram of CNN1.

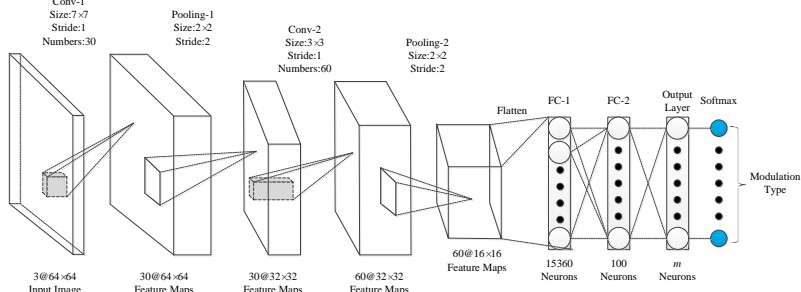

**Figure 12.** Network structure diagram of CNN2.

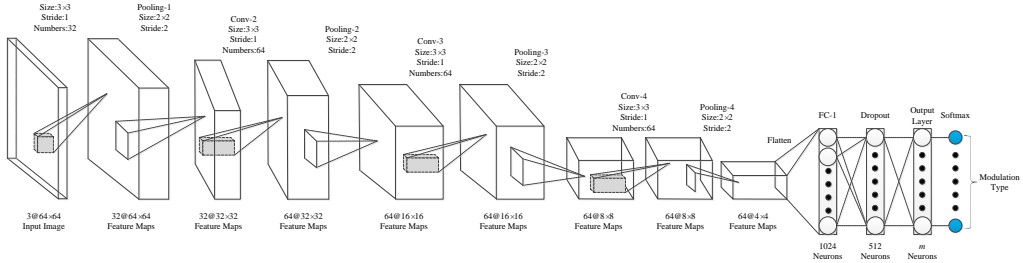

**Figure 13.** Network structure diagram of CNN3.

Let $S_{ij}^n$ be a symbol to describe *i*-th pooling layer of the *j*-th feature image in size $n \times n$; Let $C_{ik}^n$ be a symbol to describe *i*-th convolution layer of the *j*-th feature image in size $n \times n$; Let $W_{ikj}^l$ be a symbol to describe the *j*-th convolution kernel of the *k*-th group of convolution kernel in *i*-th convolution layer, the convolution kernel size is $l \times l$, accordingly. Since the described image in the RGB mode, each input image has three channels. So, when $i = 0$, then the value of *j* can be 1, 2, and 3.

Convolution layers and pooling layers are operated alternatively when classifying the image. In the pooling operation, the *k*-th feature image of the $(i + 1)$-th pooling layer can be expressed as follows:

$$C_{(i+1)k}^n = \sum_j (S_{ij}^n \otimes W_{(i+1)kj}^l) + b_k \tag{22}$$

where, "$\otimes$" represents the convolution operation. $b_k$ is a bias of the *k*-th convolution kernel.

Then, the $(i + 1)$-th convolution layer goes into the pooling operation:

$$S_{(i+2)k}^{n/2} = Pooling(C_{(i+1)k}^n) \tag{23}$$

where, $Pooling(\bullet)$ is a pooling function of CNN which can do pooling operation with the $n \times n$ feature image in the pooling layer.

In this manuscript, the size of the pooled size is defined as $2 \times 2$, the pooling mode is the maximum pooling. After maximum pooling, the number of feature maps remains the same, but the size shrinks into $n/2 \times n/2$.

After a successive of convolution and pooling operations, a series of feature maps which is converted into a unidimensional vector will be put into a fully connection network to form classification results as follows:

$$[y_1, y_2, \ldots, y_{m-1}, y_m] = Fullconnect(S_{ij}^n) \tag{24}$$

where, $Fullconnect(\bullet)$ represents a fully connected function.

In CNN1, it includes a two-layer convolutional layer and a two-layer pooling layer. According to the parameters of CNN1, it can be modeled as

$$\begin{cases} C_{1q_1}^{64} = \sum\limits_{q_0=1}^{3} (S_{0q_0}^{64} \otimes W_{1q_1q_0}^{5}) + b_{q_1} & S_{2q_1}^{32} = Pooling(C_{1q_1}^{64}) & q_1 = 1, 2, \ldots, 6 \\ C_{3q_2}^{32} = \sum\limits_{q_1=1}^{6} (S_{2q_1}^{32} \otimes W_{3q_2q_1}^{2}) + b_{q_2} & S_{4q_2}^{16} = Pooling(C_{3q_2}^{32}) & q_2 = 1, 2, \ldots, 16 \\ [y_1, y_2, \ldots, y_{m-1}, y_m] = Fullconnect(S_{4q_2}^{16}) \end{cases} \tag{25}$$

The model of CNN2, with two convolution layers and two pooling layers, is expressed as

$$\begin{cases} C_{1q_1}^{64} = \sum\limits_{q_0=1}^{3} (S_{0q_0}^{64} \otimes W_{1q_1q_0}^{7}) + b_{q_1} & S_{2q_1}^{32} = Pooling(C_{1q_1}^{64}) & q_1 = 1, 2, \ldots, 30 \\ C_{3q_2}^{32} = \sum\limits_{q_1=1}^{30} (S_{2q_1}^{32} \otimes W_{3q_2q_1}^{3}) + b_{q_2} & S_{4q_2}^{16} = Pooling(C_{3q_2}^{32}) & q_2 = 1, 2, \ldots, 60 \\ [y_1, y_2, \ldots, y_{m-1}, y_m] = Fullconnect(S_{4q_2}^{16}) \end{cases} \tag{26}$$

The model of CNN3, with three convolution layers and three pooling layers, is expressed as

$$\begin{cases} C_{1q_1}^{64} = \sum\limits_{q_0=1}^{3} (S_{0q_0}^{64} \otimes W_{1q_1q_0}^{3}) + b_{q_1} & S_{2q_1}^{32} = Pooling(C_{1q_1}^{64}) & q_1 = 1, 2, \ldots, 32 \\ C_{3q_2}^{32} = \sum\limits_{q_1=1}^{32} (S_{2q_1}^{32} \otimes W_{3q_2q_1}^{3}) + b_{q_2} & S_{4q_2}^{16} = Pooling(C_{3q_2}^{32}) & q_2 = 1, 2, \ldots, 64 \\ C_{5q_3}^{16} = \sum\limits_{q_2=1}^{64} (S_{4q_2}^{16} \otimes W_{5q_3q_2}^{3}) + b_{q_3} & S_{6q_3}^{8} = Pooling(C_{5q_3}^{16}) & q_3 = 1, 2, \ldots, 64 \\ C_{7q_4}^{8} = \sum\limits_{q_3=1}^{64} (S_{6q_3}^{8} \otimes W_{7q_4q_3}^{3}) + b_{q_4} & S_{8q_4}^{4} = Pooling(C_{7q_4}^{8}) & q_4 = 1, 2, \ldots, 64 \\ [y_1, y_2, \ldots, y_{m-1}, y_m] = Fullconnect(S_{8q_4}^{4}) \end{cases} \tag{27}$$

We must emphasize that the number $q_0$ of channels of the input layer image varies with the algorithm, and in the input layer of the algorithm proposed in this manuscript, $q_0 = 1$ or $q_0 = 3$, while the other algorithms use $q_0 = 1$.

*4.3. Comparison of Algorithm Performance*

4.3.1. Image Restoration Degree and Image Stability Degree

According to the evaluation formula defined in Section 3.3, this manuscript first simulates the signal processing algorithms given in Table 3, and evaluates the image stability degree and image restoration degree of each algorithm. Figure 14 shows the performance of each algorithm on two indices.

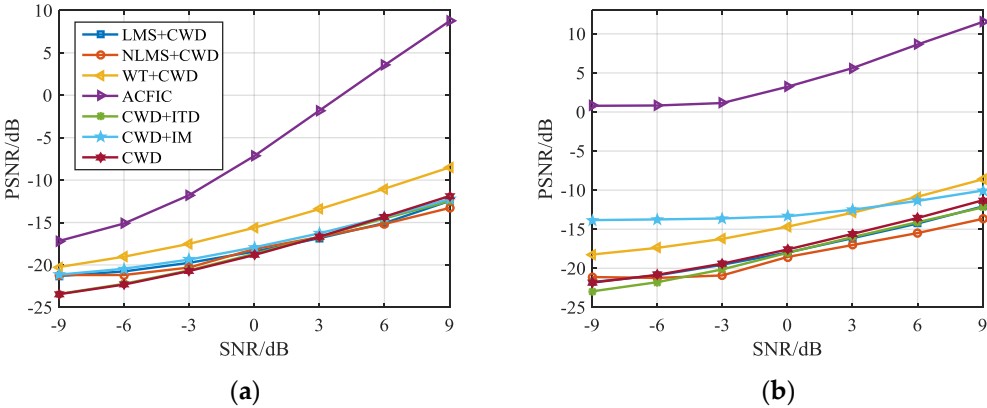

**Figure 14.** Evaluation method of the signal pre-processing algorithms: (**a**) Image restoration degree; (**b**) image stability degree.

From Figure 14, it can be seen that with the increase of the SNR, the PSNR values of each indice are improved.

The PSNR in Figure 14a was used to analyze the image restoration degree of each algorithm. Based on the definition in Section 3.3, the image restoration degree describes the difference between the feature image formed by the pre-processing algorithm with and without noise interference, so the indice can be used to evaluate the noise-suppression effect of each algorithm in different noise environments. A larger PSNR indicates better noise suppression.

In Figure 14a, we can see that WT and ACFICT have more advantages in image restoration degree than the other five algorithms. For WT and ACFICT, when SNR = –9 dB, PSNR = –21 dB in WT, and PSNR = –17 dB in ACFICT. When SNR = 9 dB, PSNR = –8 dB in WT, and PSNR = 9 dB in ACFICT. For the other five algorithms, including LMS, NLMS, CWD, and two algorithms based on ITD, when SNR rises from –9 to 9 dB, their PSNR rises from –25~–20 to –15~–10 dB. It can be found that when the SNR is –9 dB, ACFICT still outperforms the PSNR value of 4 dB than WT algorithm. Therefore, by synthesizing the above analysis, ACFICT achieves the best result in image restoration degree compared with the other six algorithms.

As shown in Figure 14b, the image stability degree was analyzed by PSNR. According to the definition of Section 3.3, image stability degree represents the difference between the two images formed by signal pre-processing algorithm under two independent noise disturbances. Therefore, this indice can be used to evaluate the stability of image features formed by the algorithm in different noise environments. The more stable each pixel characteristic of the signal is, the stronger the anti-noise ability of the algorithm is, and the bigger the PSNR value is.

Adaptive filtering (NLMS, LMS), CWD, and the method combining CWD and ITD have poor stability and share trends like that in Figure 14b. When SNR = –9 dB, then the range of their PSNR is –23~–21 dB; when SNR = 0 dB, their PSNR is –19~–17 dB; and when SNR = 9 dB, their PSNR is –14~–11 dB. WT, IM, and ACFICT have greater stability than the four algorithms above.

It can be concluded that ACFICT has greater image stability than the other six algorithms because the values of PSNR stay higher than 0 dB in the range –9~9 dB. When SNR = –9 dB, then PSNR = 1 dB, and when SNR = 9 dB, then PSNR = 11 dB. However, observing the ACFICT curve, it can be found that its PSNR value is relatively low and the change is small at a SNR of –9~–3 dB. In fact, under the SNR of –9~–3 dB, the noise almost drowns the signal, and the characteristic image composed of ACFICT gradually loses the pixel characteristics of the signal. At this time, it focuses more on reflecting the change after noise autocorrelation, so when SNR < –3 dB, its PSNR value hardly changes. When SNR > –3 dB, the feature image composed of ACFICT starts to focus more on reflecting the change after signal autocorrelation, which makes the PSNR value of ACFICT start to rise.

In summary, from the comprehensive consideration of the image restoration degree and image stability degree, it can be concluded that ACFICT is a more effective signal feature image construction algorithm.

### 4.3.2. Signal Recognition Rate

Test sets generated according to the simulation environment in Section 4.1 were used in the three networks to verify the seven pre-processing algorithms. The simulation results of the signal recognition rate are shown in Figure 15.

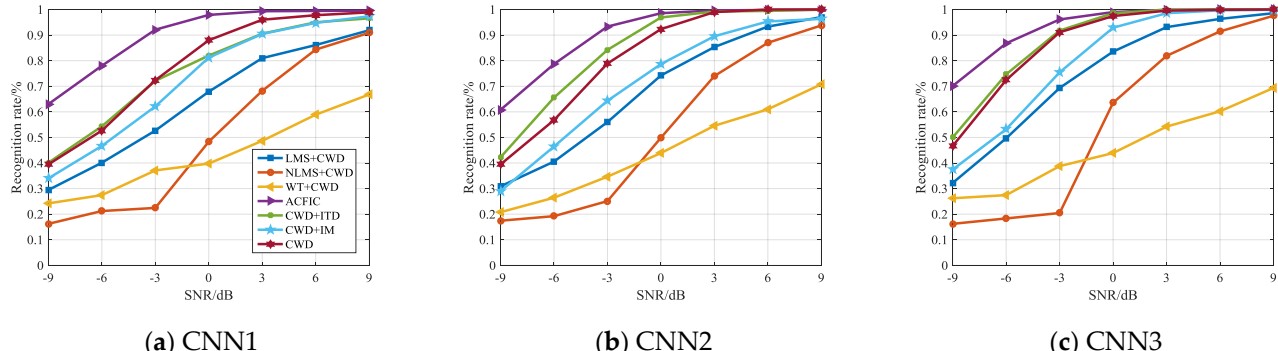

**Figure 15.** Signal recognition results of algorithms under three network models: (**a**) CNN1; (**b**) CNN2; (**c**) CNN3.

From Figure 15, it can be noted that when SNR = –9 dB, the recognition rates of ACFICT all surpass 60%, whereas ACFICT reaches 60% in CNN1 and CNN2 and 70% in CNN3, while the other six algorithms fall below 50%. In the six algorithms, CWD and the method combining CWD and ITD perform better [7,15,19], all reaching 40% in CNN1 and CNN2, and almost reaching 60% in CNN3, while the other four algorithms fall below 40%.

With the increase of SNR, there are some improvements to all the algorithms. When SNR > 3 dB, except for WT, the recognition rates of the other algorithms are higher, but the recognition rate of each algorithm has a little different in different models. ACFICT has the highest recognition rate in the simple model CNN1. ACFICT, CWD and the method combined with CWD and ITD reach approximately 100% in the CNN2 which has larger convolution kernel and more feature images. ACFICT, CWD, CWD+ITD, and CWD+ITD+IM reach approximately 100% in the CNN3 which has deeper layer.

It can be found that although the recognition rate of each algorithm is different in different models, the trend of change is the same as a whole. At the same time, under the three network models, the ACFICT proposed in this manuscript has better recognition performance, which further illustrates that the better recognition performance of ACFICT is not limited to specific network models. When we use CNN to identify signal modulation mode, the ACFICT is a better signal pre-processing algorithm.

### 4.3.3. Competitive Literature Comparison

Test sets generated according to the simulation environment in Section 4.1 were used in the comparison with the same type of competition literature. It is noted that reference [7] also studies the classification of modulation types of signals, which explores the recognition rates of random projections and sparse classification (RPSC) [29] and SCDAEs [7], and its simulation signals and research ideas are similar to this manuscript. Therefore, the recognition rate of this manuscript is further compared with that of the technology in [7]. After setting the same simulation conditions, the simulation results of the signal recognition rate are shown in Figure 16.

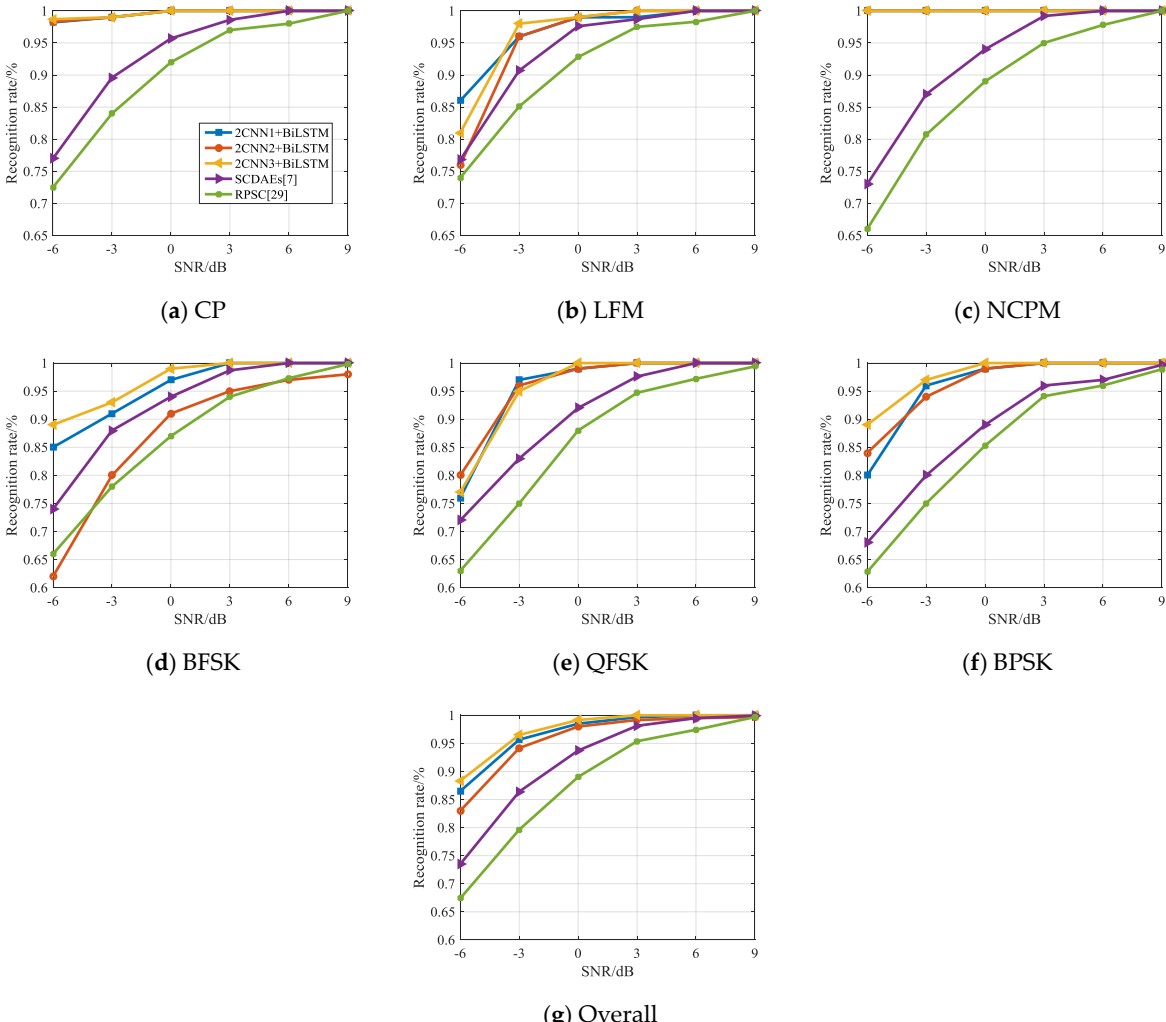

**Figure 16.** Comparison of the recognition rate between the proposed hybrid model and competitive literature.

It can be found in Figure 16 that the recognition rate of the hybrid model in CP, LFM, NCPM, QFSK, and BPSK is better than that of RPSC and SCDAEs, regardless of which CNN structure is adopted. So, from the overall perspective, the hybrid model gets a higher recognition rate.

It is noted that when the SNR is –6 dB, the recognition rate obtained by constructing the hybrid model with CNN3 is higher than that of SCDAEs and RPSC. Overall, when SNR = –6 dB, the recognition rate of CNN3 is 88%, and SCDAEs is 74%. With the increase of SNR, the recognition rate of hybrid model and SCDAEs gradually increases. However, the recognition rate of SCDAEs is comparable to that of the hybrid model only when the SNR is > 6 dB. This further illustrates that the hybrid model proposed in this manuscript has better recognition performance at low SNR on the recognition rate indicator.

It should not be overlooked that different CNN structures will affect the recognition result of the signal modulation mode, which is particularly evident in Figure 16. Among the three hybrid models, CNN3 with deeper layer has the best result, while CNN2 with large convolution kernel has the lowest recognition rate. This shows that although a better network structure has been obtained through hyper parameter adjustment in literature [11] and [20], it is still only applicable to the corresponding signal pre-processing method. In the pre-processing method proposed in this manuscript, a better network parameter configuration is shown in Figure 13.

### 4.4. Optimization Analysis of Proposing Method

4.4.1. Influence of Changing Input Layer on Signal Recognition Rate

In Section 3.2, based on ACFICT, an optimization scheme for the image of the CNN's input layer was proposed. Specifically, ACFICT was used to obtain three feature images that can represent signals, and then three images are input into three parallel deep learning models to complete the classification task. Figure 17 shows the recognition results based on a set of and three sets of parallel deep learning models. For a more intuitive comparison, we also draw the recognition results of literature [16] into them.

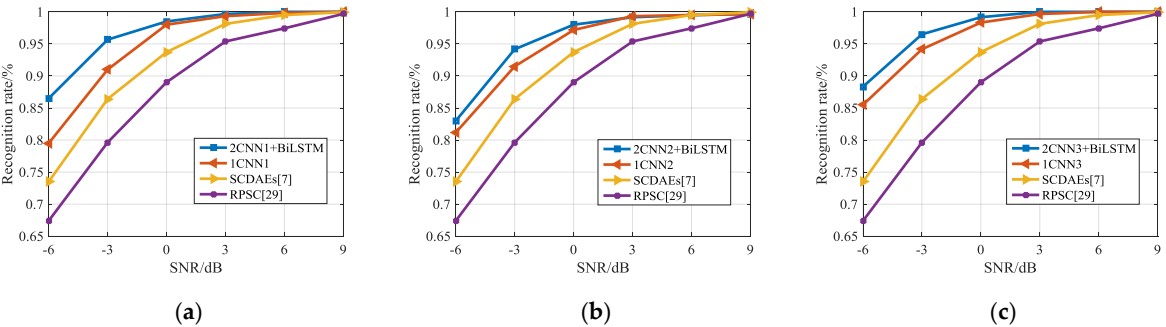

(a)　　　　　　　　(b)　　　　　　　　(c)

**Figure 17.** Analysis of the influence of changing input layer on signal recognition rate. (**a**), (**b**) and (**c**) are the recognition results obtained by constructing the hybrid model with CNN1, CNN2, and CNN3, respectively.

It can be seen from Figure 17 that based on ACFICT, the recognition result obtained by using a single set of CNN is higher than that proposed by the literature [7]. In the comparison of CNN, no matter which structure is adopted, the hybrid model can obtain better recognition rate improvement than the original CNN structure. This shows that the improvement of the recognition rate of the hybrid model is not due to a specific deep learning model, which can better integrate multiple input images to obtain better recognition results. Specifically, when CNN1 is adopted, the hybrid model increases the recognition result of the original CNN1 by 7%, and for CNN2 and CNN3, it increases by 2% and 3%, respectively. As the SNR increases, the recognition rates of CNN and hybrid models tend to be consistent at SNR = 3 dB.

4.4.2. Recognition Speed Analysis

Tables 4 and 5 show the processing speeds obtained when changing the number of input feature images and the classification model. The GPU for processing is GTX 1050Ti and the CPU is i7-8565U. It should be emphasized that the simulation results are the average values obtained after 100 Monte Carlo simulations.

**Table 4.** Signal pre-processing speed analysis.

| Number of Feature Images | Processing Speed |
| :---: | :---: |
| 1 | 0.0356 s |
| 2 | 0.0535 s |
| 3 | 0.0845 s |

**Table 5.** Speed analysis of classifier recognition.

| Classifier | Processing Speed |
| --- | --- |
| CNN1 | 0.00027 s |
| CNN2 | 0.00081 s |
| CNN3 | 0.00089 s |
| 2CNN1+BiLSTM | 0.00253 s |
| 2CNN2+BiLSTM | 0.00341 s |
| 2CNN3+BiLSTM | 0.00346 s |

It can be found that the time consumption of the method proposed in this manuscript is mainly in the signal pre-processing stage. Although more input information is introduced into multiple autocorrelation feature maps, it inevitably leads to the increase of calculation time. For each additional autocorrelation feature map, the pre-processing speed of ACFICT will increase by 0.02~0.03 s. For the recognition of the classifier, its processing speed is much faster than that of the signal pre-processing stage. However, it cannot be ignored that the high recognition results obtained by using multiple groups of deep learning models in this manuscript also lead to a reduction of one order of magnitude in the recognition speed of the classifier. In terms of the recognition speed, no matter which CNN structure is used, the hybrid model is 10 times lower than the single group CNN. Therefore, according to the method proposed in this manuscript, a conclusion reflecting the simulation results of this manuscript is that ACFICT proposed in this manuscript is an effective signal pre-processing algorithm, which provides a new feature image for signal classification under low SNR, and further improves the signal recognition rate through the hybrid model. However, the average time obtained in signal pre-processing and classifier recognition reveals that ACFICT and hybrid model bear more computing load. Increasing the number of autocorrelation feature images and parallel deep learning models in hybrid model will reduce the recognition speed of the proposed method.

## 5. Conclusions

ACFICT was proposed to construct signal feature images that are robust enough to represent signals at low SNR. According to the results of deviation analysis, we propose to combine the autocorrelation sequences with TFA in a specified range to form TFI. As for the decrease of pixel intensity of feature images, which may be caused by SNR reduction, we further present a multiple-autocorrelation method to overcome the influence of low-SNR signal feature images and proposed the optimization scheme of an input-layer image based on multiple-autocorrelation theory. In the testing stage, ACFICT is compared with six types of signal pre-processing algorithms on three indices, which are image restoration degree, image stability degree, and signal recognition rate. At the same time, the input layer optimization scheme based on ACFICT can further improve the signal recognition rate at low SNR. Three images that generated by ACFICT are input into three parallel deep learning models called hybrid model. In the comparison of CNN, no matter which structure is adopted, the hybrid model can obtain better recognition rate improvement than the original CNN structure. This shows that the improvement of the recognition rate of the hybrid model is not due to a specific deep learning model, which can better integrate multiple input images to obtain better recognition results. In comparison with the competition literatures, the simulation results further prove that ACFICT combined with the hybrid model is a more effective signal modulation type classification method. When the SNR is –6 dB, the overall recognition rate of the method reaches 88%. So, the hybrid model combined with the ACFICT can maintain the unique feature of each signal and is not susceptible to noise interference at the low-SNR environment, which meets the requirements of signal classification and is easier to classify signals.

**Author Contributions:** Guidance of theoretical analysis and writing, Z.M. and Z.H.; Operation of the experiments, analysis, and writing of the manuscript, Z.M.; Guidance of theoretical analysis, G.H.; Guidance and optimizing of experiments, A.L.

**Funding:** This work was supported in part by the National Natural Science Foundation of China under Grant 61501484.

**Conflicts of Interest:** The authors declare no conflict of interest.

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
