# Peer review of "Emitter Signal Waveform Classification Based on Autocorrelation and Time-Frequency Analysis"

_electronics, doi:10.3390/electronics8121419_

Round 1

Reviewer 1 Report

The article is based on the previous one, entitled “A Novel Feature Image Construction Algorithm for Radar Waveform Recognition.” Besides the change of terminology (radar – emitter), the significant differences are as follows:

A few figures are left out from the manuscript (e.g., Fir. 4 b, d, and f), etc. Section 3.1.5 is reduced Section 3.2 has been slightly completed Section 4.3.3 has been added Section 4.4. has been completed Conclusions are slightly changed Other minor changes and improvements

Six types of signals are described in the article (chapter 3.1.1). These are used to verify the authors’ algorithms – ACFICT. Is it possible to use a real sample of a real signal, including the potential noise, multipath propagation, etc…?

Overall, due to my opinion, the article might be interesting to readers, and I recommend it to publish.

Author Response

Point 1:  A few figures are left out from the manuscript (e.g., Fig. 4 b, d, and f), etc. Section 3.1.5 is reduced Section 3.2 has been slightly completed Section 4.3.3 has been added Section 4.4. has been completed Conclusions are slightly changed Other minor changes and improvements 

Response 1: Modified according to reviewer's comments for Fig 2, Fig 3, Fig4, Fig 5 and Fig 6. (lines 162-165, 192-195, 235-236, 249-250, 270-271).When reviewed the figures, we found that the original figures were wrong in Figure 2. The figures we originally wanted to present are the six types of signals simulated with an SNR of 9 dB in this manuscript. However, the original figures are the six types of signals simulated with noise free environment. We have modified and further compared with figure 3. (lines 192 - 194). In addition, the structure of the article was slightly adjusted about Section 4.2.2. (lines 445-463)

Point 2: Six types of signals are described in the article (chapter 3.1.1). These are used to verify the authors’ algorithms – ACFICT. Is it possible to use a real sample of a real signal, including the potential noise, multipath propagation, etc…?

Response 2: The reviewer's comments are very forward-looking. The next step that our team plans to do is to verify the actual algorithm. We will design many factors that influence the actual condition, because signals generated by different types of electromagnetic sources are in many situations noisy, misshaped or changing in relation to the weather condition, task and application. In this case, we will use the software radio system to sample the actual data to verify the algorithm proposed in our study.

Reviewer 2 Report

REVIEW

Article titled “Emitter Signal Waveform Classification Based on Autocorrelation and Time-frequency Analysis”

Electronics no:  641928

List of Authors: 

Zhiyuan Ma, Zhi Huang, Anni Lin, and Guangming Huang

In this paper the Authors proposed an autocorrelation feature image construction technique combined with a convolutional neural network (CNN) for emitter waveform classification and a structure optimization for CNN input layer.

The Authors in Introduction (29-80) have reviewed of world literature concerning the most important aspects of signal waveform recognition techniques, applied models of neural network structures, the ways of reducing the number of parameters and computational complexity in images classification and recognition. The problem of emitter signal classification is very important in environment monitoring where signals generated by different types of electromagnetic sources (emitters, radars) are in many situations noisy, misshaped or changing in relation to the weather condition, task and application. Presented by the Authors a hybrid model of emitter signal waveform classification with using the convolutional neural networks may also be directly applicated in electronic warfare systems for emitter recognition and identification, thus the following articles concerning the similar problem are also supposed to be listed in the References: Neural network application for emitter identification. (2017) Proceedings International Radar Symposium, 28-30 June 2017, Prague, Czech Republic, art. no. 8008202, DOI: 10.23919/IRS.2017.8008202. Adaptive Forming the Beam Pattern of Microstrip Antenna with the Use of an Artificial Neural Network. International Journal of Antennas and Propagation, Hindawi Publishing Corporation, Volume 2012, Article ID 935073, doi:10.1155/2012/935073 (ISSN: 1687-5869). Radar signal identification using a neural network and pattern recognition methods. (2018) 14th International Conference on Advanced Trends in Radioelectronics, Telecommunications and Computer Engineering, TCSET 2018 - Proceedings, 2018-April, pp. 79-83. DOI: 10.1109/TCSET.2018.8336160. Recognition of electromagnetic sources with use of deep neural networks. 12th Conference on Reconnaissance and Electronic Warfare Systems (CREWS), Oltarzew, Poland, 19-21.11.2018, Proc. SPIE 11055, 110550D (27 March 2019); DOI: 10.1117/12.2524536. The utilization of unintentional radiation for identification of the radiation sources. 34 European Microwave Conference EuMC 2004, Amsterdam, The Netherlands 12-14 October 2004, vol. 2, pp. 777-780.

The successive steps of proposed method and way of solution are correctly presented with the sufficient details in Sections 3 and 4. Additionally, in understanding the proposed method are very useful illustrations shown in Figures 1 to 14. The experiment results based on simulated data are properly illustrated in Figures 15 to 18 and in Tables 4 and 5. The obtained results for proposed hybrid model in comparison with competitive literature algorithms give a higher recognition rate. The proposed method with appropriate formulas, equations and symbols are correctly Additionally, for understanding the proposed method and obtained calculation results are very useful illustrations depicted in Figures 15-18 and Tables 4 and 5. The proposed by Authors the hybrid model of emitter signal waveform classification based on autocorrelation and time-frequency analysis gives the better performance and recognition improvement than the original CNN structure what prove its utility.

Remarks:

The word “where” should be written in a small letter after equation (lines 147, 149,157, 171,176, 202, 276) because it is the further part of sentence. Not all abbreviations are deciphered, for example:

ECG – line 77, BiLSTM – line 94, IF – line 112, GPU and CPU – line 121, NCFM – in Table1, PSNR – line 400, RPSC – line 560, SCDAE – line 561.

The authors use two different concepts: the mean square deviation (Eq. 18) and the mean square error (Eq. 28). Do they mean the same concept of error? Could the Authors provide for what values of time and frequency are illustrated the images in Figures 3, 4, 7, 8 and 9. The axis (horizontal and vertical) are not described. Based on the calculation made could the Authors give few sentences in Conclusions about possibility of recognition the different type of emitter signal mode.

The paper correctly describes the problem of emitter signal waveform recognition and classification, the way of solution and correctly presents the results of computational experiments achieved for the simulated data. The obtained research results may be directly applied in electronic warfare systems for emitter signals identification or classification.

The Authors consider the problem and proposed its solution which is relevant and interesting for publication after making these pointed corrections.

However, I cannot recommend the current manuscript for publication unless the current version is revised. After providing the answers to the questions above, the work is supposed to be reviewed once again.

Author Response

Point 0: Some articles concerning the similar problem are also supposed to be listed in the References. 

Response 0: We have study these papers seriously and really feel that we have benefited a lot. We have listed them in the References and added information that we needed in this manuscript. (lines 43-48, lines 644-648, lines 655-658, References [4], [5], [9], [10])

Point 1: The word “where” should be written in a small letter after equation (lines 147, 149,157, 171,176, 202, 276) because it is the further part of sentence. 

Response 1: Modified according to reviewer's comments. (lines 139, 144, 158, 181, 187, 212, 230, 453, 455, 463)

Point 2: 2. Not all abbreviations are deciphered, for example:

ECG –line 77, BiLSTM –line 94, IF –line 112, GPU and CPU –line 121, NCFM –in Table1,

PSNR –line 400, RPSC –line 560, SCDAE –line 561.

Response 2: Modified according to reviewer's comments. Including ECG, BiLSTM, IF, GPU, CPU, PSNR, RPSC, SCDAE. (lines 79, 96, 113, 122, 122, 351, 544, 60)

NCFM is a spelling mistake, has changed NCFM to NCPM. (lines 147, 151, 282, 284, 548, 550)

Point 3: The authors use two different concepts: the mean square deviation (Eq. 18) and the mean square error (Eq. 28). Do they mean the same concept of error?

Response 3: They are two different concepts.

Square deviation: Analysis for error between the autocorrelation value within a given time T range and the theoretical value of autocorrelation.

Square error: Indicator for quality of images that characterize different signals.

Point 4: Could the Authors provide for what values of time and frequency are illustrated the images in Figures 3, 4, 7, 8 and 9. The axis (horizontal and vertical) are not described.

Response 4: Modified according to reviewer's comments. Added the axis (horizontal and vertical) and all the figures have been completed. (Fig 2, Fig 3, Fig 4, Fig5, Fig6, Fig 7, Fig 8), (lines 162-165, 192-195, 235-236, 249-250, 270-271, 286, 305).

Point 5: Based on the calculation made could the Authors give few sentences in Conclusions about possibility of recognition the different type of emitter signal mode.

Response 5: Modified according to reviewer's comments. Have already add in Abstract and Conclusions. (lines 21-23, lines 626-628)

Round 2

Reviewer 2 Report

REVIEW_2

Article titled “Emitter Signal Waveform Classification Based on Autocorrelation and Time-frequency Analysis”

Electronics no:  641928

List of Authors: 

Zhiyuan Ma, Zhi Huang, Anni Lin, and Guangming Huang

The article Electronics no. 641928 entitled “Emitter Signal Waveform Classification Based on Autocorrelation and Time-frequency Analysis” has been carefully modified and well revised. The work is supposed to be finally accepted for publication in Electronics.

This manuscript is a resubmission of an earlier submission. The following is a list of the peer review reports and author responses from that submission.

Round 1

Reviewer 1 Report

First of all, I thank the authors for their work. Despite the careful research, the formal aspect of the article is underestimated. Some images are missing a description/legend in English.
Regarding the formal side of the review, I recommend the authors:
- Review pictures and add descriptions and legends in English
- Check the axis descriptions in the figures. Many of them are missing (for example, in Fig. 6, 9, 10, etc.).

Based on the "professional" part of the review, I have the following questions and comments:
What indicates the color range used in Fig. 3 (and the subsequent similar figures)? Please complete.

A signal generator was used to verify the proposed algorithm and to compare it with the other six algorithms. Why did the authors not use a real recording of a real signal? In my opinion, this element would lead to a substantial increase in the value of the article.

Based on the previous question, the next one arises. What are the real-time applications of technology? Will computational complexity allow it to be used directly on the battlefield? Is there any comparison to other methods available?

Reviewer 2 Report

The  real problems of Radar Recognition are much different from those considered in this paper. A main Reference, which shall better substitute [1] of this paper, could be : [1'] A. De Martino, "Introduction to EW systems", Artech House, 2nd Edition, 2012, ISBN 978 1 60807 201 1 . Also, with respect to [2], in the literature there is much more and better.

The material presented in this paper has little to do with its title, and has a very limited interest.

Concerning formal aspects, some terms leave the reader very surprised, such as "pixel feature" , firstly found in row 7 of the Abstract, and "Two-dimensional time-frequency images"  firstly found in the Introduction, row 9, and the whole rows 100 to 108 , page 2, section 2.

Concerning formal aspects, some terms leave the reader very surprised, such as "pixel feature" , firstly found in row 7 of the Abstract, and "Two-dimensional time-frequency images"  firstly found in the Introduction, row 9, and the whole rows 100 to 108 , page 2, section 2.

The fact that  both frequency spectrum  (central frequency,  bandwidth) and duration of the signal to be intercepted are a priori unknown is completely forgotten in the proposed  approach.

The selected waveforms in Table 1 are all with high sidelobes in their autocorrelation function, so, hardly used in any practical radar system.